# Impact of Serious Games on Body Composition, Physical Activity, and Dietary Change in Children and Adolescents: A Systematic Review and Meta-Analysis of Randomized Controlled Trials

**DOI:** 10.3390/nu16091290

**Published:** 2024-04-26

**Authors:** Mingchang Liu, Xinyue Guan, Xueqing Guo, Yixuan He, Zeqi Liu, Shiguang Ni, You Wu

**Affiliations:** 1Institute for Hospital Management, School of Medicine, Tsinghua University, Beijing 100084, China; liumc22@mails.tsinghua.edu.cn (M.L.); guanxy22@mails.tsinghua.edu.cn (X.G.); liuzq21@mails.tsinghua.edu.cn (Z.L.); 2Sargent College of Health & Rehabilitation Sciences, Boston University, Boston, MA 02215, USA; xqguo@bu.edu; 3Department of Bioengineering, East China University of Science and Technology, Shanghai 200237, China; 20000896@mail.ecust.edu.cn; 4Shenzhen International Graduate School, Tsinghua University, Shenzhen 518055, China; ni.shiguang@sz.tsinghua.edu.cn; 5Department of Health Policy and Management, Bloomberg School of Public Health, Johns Hopkins University, Baltimore, MD 21205, USA

**Keywords:** children, adolescents, serious game, body composition, physical activity, dietary change

## Abstract

Over the past four decades, obesity in children of all ages has increased worldwide, which has intensified the search for innovative intervention strategies. Serious games, a youth-friendly form of intervention designed with educational or behavioral goals, are emerging as a potential solution to this health challenge. To analyze the effectiveness of serious games in improving body composition, physical activity, and dietary change, we performed a systematic review and meta-analysis of randomized controlled trials (RCTs) from PubMed, Web of Science, EMBASE, and Scopus databases. Pooled standardized mean differences (SMD) were calculated for 20 studies (*n* = 2238 the intervention group; n = 1983 in the control group) using random-effect models. The intervention group demonstrated a slightly better, although non-significant, body composition score, with a pooled SMD of −0.26 (95% CI: −0.61 to 0.09). The pooled effect tends to be stronger with longer duration of intervention (−0.40 [95% CI: −0.96, 0.16] for >3 months vs. −0.02 [95% CI: −0.33, 0.30] for ≤3 months), although the difference was not statistically significant (*p*-difference = 0.24). As for the specific pathways leading to better weight control, improvements in dietary habits due to serious game interventions were not significant, while a direct positive effect of serious games on increasing physical activity was observed (pooled SMD = 0.61 [95% CI: 0.04 to 1.19]). While the impact of serious game interventions on body composition and dietary changes is limited, their effectiveness in increasing physical activity is notable. Serious games show potential as tools for overweight/obesity control among children and adolescents but may require longer intervention to sustain its effect.

## 1. Introduction

Childhood obesity has a profound impact on both physical and mental health, resulting in nutritional deficiency and a higher risk of developing diseases such as diabetes and cardiovascular disease [1]. In addition, overweight and obese children are more likely to suffer from depression and other psycho-socio problems that lead to a significant decrease in their quality of life. However, the prevalence of childhood obesity continues to increase due to unhealthy modern lifestyles. Between 2017–2020, the prevalence of obesity reached 19.7% and affected approximately 14.7 million youth at the age of 2–19 [2], indicating the need for effective interventions.

The WHO recognizes low physical activity as one of the major contributors to poor health outcomes, and very few children manage to meet the recommended daily intake of fruits and vegetables [3,4]. Traditional intervention programs thus focus on stimulating behavioral change through a healthy diet and physical activity. However, several factors, including financial costs, lack of referrals from healthcare professionals, program availability, patient embarrassment, missing meetings, etc., have been shown to have a negative impact on the sustainability of such traditional approaches [5]. To address the issue, serious games, a type of gamified intervention, are designed with the aim to promote health literacy on nutritional knowledge and encourage physical activity among the youth [6]. Serious games incorporate the use of technologies such as mobile phones and the internet as well as game mechanics into the learning process [7]. Within this context, serious games targeting physical activity are specifically designed to involve children in movement-based gameplay, encouraging them to be more active, whereas those aimed at dietary changes gamify nutritional education, making learning about healthy eating habits both interactive and enjoyable for the youth [8,9]. More specifically, the intrinsic characteristics of serious games often involve incentives or rewards that are shown to be effective in stimulating motivation among teenagers [7,10].

Yet, there are not many reviews that evaluate the effectiveness of a serious game on weight control among the youth, and even fewer have quantified its intervention effect on both the result of weight control and the underlying behavioral changes. Variations in study population, intervention methods, and outcome measures make it harder to quantify the overall effect. Previous meta-analyses on this topic have limitations that justify further investigation: Some employed incomplete search strategies and examined the effect based on only two studies [11], some were not able to summarize the results quantitatively [12,13], others overlooked newly developed tools that emerged over recent years [13,14], and several failed to explore the behavioral changes facilitated by serious game interventions [15].

This study evaluates the effectiveness of serious games in improving body composition, physical activity, and dietary habits among children and adolescents. To overcome the limitations of previous studies, the primary outcomes of our study include a range of comprehensive measures of body composition (including body mass index z-score (BMI z-score), body weight, and percentage of fat mass). To enhance the understanding of how gamified interventions achieve their effectiveness, this study takes an additional step beyond previous research by evaluating physical activity and dietary change, where feasible, as intermediary markers. Our hypothesis is that serious games can lead to short- to mid-term changes in health-related behaviors, such as increased exercise and improved dietary habits among adolescents. If these changes are sustained, they could significantly contribute to long-term obesity control.

## 2. Materials and Methods

This study was registered in the International Prospective Register of Systematic Reviews (PROSPERO, registration number: CRD42023425612). The study followed the Preferred Reporting Items for Systematic Reviews and Meta-analyses (PRISMA) reporting guideline for meta-analyses [16] (see Appendix A).

### 2.1. Information Sources and Search Strategy

We performed a record search of electronic databases including PubMed, Web of Science, EMBASE, and Scopus, with no limitations on publication type, from database inception to 29 April 2023. The search terms were a combination of words associated with children (Adolescen* OR children), game (game OR gaming OR exergam* OR serious game OR video game OR videogame OR game-based OR gamifi* OR Exergaming OR Games, Experimental OR Gamification), behavior changes (physical activity OR sport OR exercise OR eating OR food OR dietary habit* OR nutrition* knowledge OR Food Preferences OR Feeding Behavior OR Diet), body composition (BMI OR weight OR Body Mass Index OR Body Weight), and randomized controlled trial (randomized controlled trial OR controlled clinical trial OR RCT OR randomized OR trial OR randomly OR Clinical Trials as Topic). Full search details for each database are listed (see Appendix A).

### 2.2. Selection Criteria

The PICOS (Population, Intervention, Comparison, Outcome, and Study design) method was used to define the inclusion criteria for this meta-analysis and systematic review: (P) adolescents and children; (I) trials that study the effects of digital games; (C) compared with either standard education or no specific education; (O) BMI/weight, physical activity, dietary habit, food preference, etc.; and (S) randomized controlled trials. The exclusion criteria were (1) articles not in English; (2) research articles that were not journal-original; (3) irrelevant subjects; and (4) irrelevant outcome.

The results of the search were imported into Endnote (version 20.5), where duplicates were removed. Two reviewers (M.L. and Xinyue Guan) completed title/abstract screening. Assessment of full-text articles to determine eligibility of the remaining studies was conducted by the same two reviewers. Disagreements between reviewers were resolved by another researcher (Y.W.). The study selection process is summarized in Figure 1.

### 2.3. Data Abstraction

Relevant data were independently extracted and cross-checked by four researchers (Y.H., Xueqing Guo, M.L., and Xinyue Guan) using an Excel spreadsheet (Microsoft Corporation, Redmond, WA, USA). The summary statistics collected for each outcome included number of participants, means, and standard deviation (SD)s. Other information extracted from each RCT included the name of the first author, country of origin, year of publication, sample size, characteristics of participants, intervention, comparison, and outcome.

### 2.4. Data Analysis

#### 2.4.1. Quality Assessment

The risk of bias in the trials included in this study was evaluated by RoB 2 (22 August 2019 version) for individually randomized trials, RoB 2 for cluster-randomized trials (18 March 2021 version), and RoB 2 for crossover trials (18 March 2021 version). The domains of the RoB 2 tool for individually randomized trials are shown in Figure 2. Two investigators assessed the risk of bias in the included studies, with differences resolved by mutual assent, before reaching a final decision with the support of a third investigator (Z.L.).

#### 2.4.2. Statistical Analysis

Due to the inherent heterogeneity across studies caused by different gaming genres and different intervention duration, we performed meta-analysis by random-effect model for body composition metrics, physical activity, and dietary change. Subgroup meta-analysis was conducted for body composition by the participants’ baseline weight (overweight/obese vs. normal weight) as well as by the duration of the intervention (>3 months vs. ≤3 months). Standardized means differences (SMDs) were calculated as the mean change from pregame to postgame in the intervention group minus the mean change from pregame to postgame in the control group divided by the standard deviation (SD). Higgins *I*^2^ was used to measure heterogeneity, with 25% representing low heterogeneity, 50% representing moderate heterogeneity, and >75% representing high heterogeneity [17]. A leave-one-out sensitivity analysis was applied for each meta-analysis. Funnel plots were used to assess publication bias when more than 10 studies were included in a specific meta-analysis [18,19]. Data analysis was performed using R (version 4.3.1) package **meta**.

## 3. Results

### 3.1. Study Selection

A total of 3468 articles were identified through database searches. After deleting 1438 duplicates, the titles and abstracts were reviewed for the remaining 2030 articles. Then, 1871 articles were excluded due to their deemed inapplicability to the research question. The full texts of 148 articles were retrieved. Articles were further excluded due to sole focus on the immediate effect of games (20), wrong control group (40), irrelevance to weight/behavioral improvement (14), or absence of extractable data (48). The current review consists of 25 articles (Figure 1).

### 3.2. Quality of Evidence

A low risk of bias in the randomization process domain was observed in 40% (10/25) of the studies. In 16 (64%) of the 25 studies, the risk of bias in the deviations from the intended interventions was low. The majority of the studies were judged to have a low risk of bias in three domains: missing outcome data (23/25, 92%), measurement of the outcome (22/25, 88%), and selection of the reported result (20/25, 80%). According to these judgments, 5 (20%) of the 25 studies were judged to have a high risk of bias in the overall bias; therefore, only 20 articles were included in the meta-analysis. The risk of bias of the studies are shown in Figure 2.

### 3.3. Study Characteristics

The 20 articles included 8844 participants from a variety of countries, including the United States (*n* = 10), New Zealand (*n* = 2), the United Kingdom (*n* = 2), Australia (*n* = 1), Canada (*n* = 1), Finland (*n* = 1), Italy (*n* = 1), the Netherlands (*n* = 1), and Portugal (*n* = 1). The sample size ranged from 28 to 3110, with the duration of intervention ranging from 2 weeks to 28 weeks. The characteristics of each study are summarized in Table 1.

### 3.4. Effects of Serious Games on Body Composition

Ten studies provided data on body composition metrics, with a final sample of *n* = 2238 subjects in the intervention group and *n* = 1983 in the control group. Among the 10 studies, 6 focused on overweight children, while 4 targeted non-overweight children. Interventions were consistently implemented for a duration of at least one month, with outcome measures encompassing BMI z-score [20,21,22], body fat [22,23], and weight [21,24,25]. Across various studies, the impact of serious games on the body composition of children exhibited variability. Some studies suggested that serious games can reduce body fat in children [20,26,27], while one study indicated no significant change in body fat among children before and after the intervention [23]. Several studies proposed a significant reduction in children’s BMI/BMI z-score with the implementation of serious games [21,22,27], yet others contended that the influence of serious games on children’s BMI/BMI z-score is not statistically significant [20,28,29]. The pooled SMD between groups was −0.26 (95% CI −0.61 to 0.09), indicating a reduced level of obesity reflected by the body composition measures. The heterogeneity of the meta-analysis, however, was high across the studies (*I*^2^ = 83%; *p* < 0.01; Figure 3).

A subgroup analysis was carried out to check whether serious games have a different effect on the body composition of overweight vs. normal weighted children and adolescents. The subgroup analysis showed that serious games had a larger effect size on the overweight children (SMD −0.35, 95% CI −0.95 to −0.26) than the non-overweight children (SMD −0.18, 95% CI −0.45 to 0.10), but the difference was not statistically significant (*p* = 0.61; Figure 4).

When stratified by the time of intervention, the effect of serious games appeared stronger in studies with a longer duration of intervention (>3 months; SMD = −0.40, 95% CI −0.96 to 0.16) compared with those with a shorter duration (≤3 months; SMD = −0.02, 95% CI −0.33 to 0.30), although the difference was not statistically significant (*p* = 0.24; Figure 5).

When comparing the mean age of the participants, serious games had a greater effect size for children under 14 years old (SMD −0.40, 95% CI −1.13 to 0.33) than those who were 14 years and above (SMD −0.01, 95% CI −0.17 to 0.15). However, the difference in the effect size between the groups was not statistically significant (*p* = 0.31, Figure 6).

### 3.5. Effects of Serious Games on Physical Activity

Among 10 studies with measured physical activity, 489 subjects were in the intervention group and 425 in the control group. The durations of interventions ranged from 4 to 28 weeks. The majority of studies indicated a significant increase in physical activity with the implementation of serious games [21,29,30,31,32,33,34], while some research did not [23,35,36]. The meta-analysis results show a significant effect of serious games in this outcome (SMD = 0.61, 95% CI 0.04 to 1.19). These results indicate that serious games enhance the level of physical activity in children. The heterogeneity of the meta-analysis was high (*I*^2^ = 92%; *P*-heterogeneity < 0.01; Figure 7).

### 3.6. Effects of Serious Games on Dietary Change

The effects of serious game and no or sham intervention on dietary change were compared in 5 of the 20 studies. These five studies involved a total of 1796 intervention group samples and 1913 control group samples. The outcome measured in the five studies included sugar intake [36], snack consumption [34], and vegetable and fruit consumption [37,38,39]. These studies indicated a significant reduction in children’s consumption of sugar [36] and snacks [34] through the implementation of serious games. Children engaging in serious games demonstrated an increased intake of vegetables and fruits, with this positive effect being statistically significant in two studies [36,39] but not in another [37]. As shown in Figure 7, a meta-analysis of the results of these studies showed a nonsignificant difference (SMD = 0.05, 95% CI −0.03 to 0.13). The statistical heterogeneity of the evidence was low (*I*^2^ = 4%; *P*-heterogeneity = 0.38, Figure 8).

**Table 1 nutrients-16-01290-t001:** Study characteristics.

Author (Year)	Country	Sample	Intervention	Duration	Measurement
A.E. Staiano (2015) [26]	United States	N = 41 Age 14–18 years n IG = 22 n CG = 19	IG: a group-danced-based exergame program CG: no intervention	12 weeks	BMI z-score
Ralph Maddison (2011) [20]	New Zealand	N = 322 Age 10–14 years n IG = 160 n CG = 162	IG: PlayStation Eye Toy (Sony), an upgrade of existing gaming technology CG: no intervention	24 weeks	BMI z-score
Stewart G. Trost (2014) [21]	United States	N = 75 Age 8–12 years n IG = 34 n CG = 41	IG: program and active gaming intervention CG: program-only intervention	16 weeks	BMI z-score; MVPA
T.L. Wagener (2011) [28]	United States	N = 40 Age 12–18 years n IG = 21 n CG = 19	IG: a group dance-based exergame exercise program CG: no intervention	10 weeks	BMI z-score
Cristina Comeras-Chueca (2022) [27]	United States	N = 28 Age 9–11 years n IG = 20 n CG = 8	IG: an active video games exercise program combined with multicomponent exercise CG: daily activities without modification	5 months	BMI z-score; MVPA
A.E. Staiano (2013) [24]	United States	N = 54 Age 15–19 years n Cooperative group = 19 n Competitive group = 19 n Control group = 16	All exergame participants were encouraged to play the Nintendo Wii Active game IG1: cooperative exergame IG2: competitive exergame CG: no intervention	20 weeks	Body weight
Lee E.F. Graves (2010) [23]	United Kingdom	N = 42 Age 8–10 years n IG = 22 n CG = 20	IG: received two jOG devices to use at home with the purpose to reduce sedentary time CG: received no device and continued with normal activity	12 weeks	Body fat; TPA
Riitta Pyky (2017) [25]	Finland	N = 496 Mean age 17.8 years n IG = 250 n CG = 246	IG: received a mobile service (MOPOrtal) to motivate participants physically, mentally, and socially CG: did not receive the mobile service	6 months	Body weight
Ann E. Maloney (2008) [29]	United States	N = 60 Age 7–8 years n IG = 40 n CG = 20	IG: a video game (DDR) intervention CG: no intervention	Intervention: 10 weeks; Follow-up: 28 weeks	BMI z-score; VPA
Viggiano, A. (2015) [22]	Italy	N = 3110 Age 9–19 years n IG = 1663 n CG = 1447	IG: involved in 15–30-min-long game sessions in class with the board game Kaledo CG: no intervention	Intervention: 20 weeks; Follow-up: 6 months, 18 months	BMI z-score
Artur Direito (2015) [35]	New Zealand	N = 51 Age 14–17 years n IG1 = 17 n IG2 = 16 n CG = 18	IG1: use of an immersive app (Zombies, Run!) with game themes IG2: use of a non-immersive app (Get Running-Couch) CG: no intervention	Intervention: 8 weeks; Follow-up: 8 weeks, 3 months	MVPA
Avril Johnstone (2019) [30]	United Kingdom	N = 137 Age 7 years n IG = 73 n CG = 64	IG: active play (formally known as Go2Play Active Play) intervention CG: no intervention	Intervention: 10 weeks; Follow-up: 9 weeks	Time spent in MVPA
Sadye Paez Errickson (2012) [31]	United States	N = 60 Age 7–8 years n IG1 = 18 n IG2 = 22 n CG = 20	IG1: a DDR (Dance Dance Revolution intervention without coaching IG2: a DDR (Dance Dance Revolution) intervention with coaching CG: no intervention	10 weeks	VPA
Andrew Miller (2015) [32]	Australia	N = 168 Age 10–12 years n IG = 97 n CG = 71	IG: a Professional Learning for Understanding Games Education (PLUNGE) program CG: no intervention	Intervention: 7 weeks; Follow-up: 8 weeks	In-class PA
Shreela V. Sharma (2015) [36]	United States	N = 107 Age 9–11 years n IG = 53 n CG = 54	IG: a Quest to Lava Mountain (QTLM) computer game intervention which was incorporated into the school curriculum CG: no intervention	6 weeks	No. of days per week involved in 30 min exercise; fruit and vegetables consumption
Ainara Garde (2015) [33]	Canada	N = 42 Age 9–13 years	Mobile Kids Monster Manor (MKMM), an exergame which participants were encouraged to play during school time Participants were randomly assigned into one of the two groups, one of which received the intervention in week 2 and the other in week 4	4 weeks	PA
Jorinde Spook (2016) [34]	Netherlands	N = 501 Mean age 17.28 n IG = 250 n CG = 251	IG: a serious self-regulation game intervention targeting students’ overweight-related behaviors named “Balance It” CG: no intervention	Intervention: 4 weeks; Follow-up: 4 weeks	VPA; fruit and vegetables intake
Braga-Pontes, C. (2022) [37]	Portugal	N = 162 Age 3–6 years n IG1 = 39 n IG2 = 40 n IG3 = 46 n CG = 37	IG1: digital game IG2: storybook IG3: storybook and stickers CG: no intervention	Intervention: 5 weeks; Follow-up: 6 months	Lettuce, carrot, purple cabbage, cucumber, and tomato consumption/portions
Cullen, K.W. (2005) [38]	United States	N = 1489 Age 8–12 years n IG = 749 n CG = 740	IG: a multimedia game named Squire’s Quest! CG: no intervention	2 weeks	Fruit and regular vegetables/servings
Wengreen, H.J. (2021) [39]	United States	N = 1859 Age 5–11 years n IG = 881 n CG = 978	IG: FIT Game CG: no intervention	Intervention: 8 weeks; Follow-up: 3 months	Fruit and vegetables consumption

IG, intervention group; CG, control group; BMI, body mass index; PA, physical activity; TPA, total physical activity; VPA, vigorous physical activity; MVPA, moderate-to-vigorous physical activity.

### 3.7. Sensitivity Analysis

The sensitivity analysis results indicate that the removal of any single study in research involving body composition, physical activity, and dietary change did not significantly impact the combined outcomes, demonstrating overall robustness (see Appendix A).

### 3.8. Publication Bias

In this study, risk of publication bias was considered low for studies involving body composition and physical activity. The symmetrical distribution observed in the funnel plot (see Appendix A) supports this assessment, alongside the results of the Egger test, which indicated no significant publication bias (*p* = 0.50 for body composition; *p* = 0.38 for physical activity). However, the presence of two studies outside the funnel plot does suggest a degree of heterogeneity. Factors such as potential conflicts of interest, standardization of outcomes and analyses, and trial registration might be taken into account. The limited number of trials (<10) on dietary change precluded a robust assessment of publication bias using either the funnel plot or advanced regression-based techniques.

## 4. Discussion

This study found that serious games could positively impact body composition in children and adolescents. The subgroup analyses revealed nuanced differences: Firstly, overweight or obese children and adolescents exhibited more pronounced improvements in body composition compared to their normal-weight counterparts. Secondly, relatively long-term serious game interventions showed a more substantial change in body composition than short-term interventions. Regarding the mechanisms behind these changes, the serious game interventions were found to significantly boost physical activity levels and, to a lesser extent, encourage healthier dietary habits among the young participants.

Compared to previous studies, our research is the first systematic review and meta-analysis of RCTs to comprehensively estimate the effect of serious games on body composition, physical activity, and dietary change among children and adolescents. Some studies focused on the influence of serious games on healthy dietary, physical activity, or obesity prevention but without quantitative meta-analysis [12,13]. Some studies explored the effect of serious games on healthy lifestyle promotion regardless of age, covering healthy diet, physical activity, social behavior, health responsibility, stress management, and self-actualization [14], but the effects on children and adolescents still remain inconclusive. A. Ameryoun et al. [15] examined the effect of active games in overweight/obese children or adolescents only, while both overweight/obese and normal children or adolescents were included in our study. Suleiman-Martos et al. examined the effect of gamification on BMI among children and adolescents based on only 2 studies [11]; our analyses generated comprehensive and updated evidence based on 10 studies on body composition, which offers a more definitive summary estimate and may inspire further research in the development and application of serious games.

Our results were consistent with previous review studies, further supporting the potential use of serious games to improve body composition in children and adolescents. As noted by other authors, no significant effect of serious game was found for BMI z-score [11,28,29,40] or percentage of body fat [21,23]. The mean weight or BMI of both intervention and control group increased in some studies [25,29], possibly because (1) children and adolescents are in the growth and development period, and (2) the duration of trials was not long enough to demonstrate significant change.

The suggestive difference observed between overweight/obese vs. normal weight children and adolescents was consistent with the findings from C. Hernández-Jiménez, where a greater effect was found when the intervention was applied to overweight or obese children and adolescents [41]. It may partly be attributed to the large room for improvement at baseline. Furthermore, according to previous study, children with a higher BMI z-score had a greater increase in steps and active minutes per day [33]. Thus, another explanation is that overweight/obese children and adolescents have higher intrinsic motivation to lose weight. As noted by A. Cebolla i Martí et al., obese children scored significantly higher in expectations and satisfaction [42]. We also observed that those who has a longer duration of intervention tended to exhibit a larger effect. Having more active play sessions and/or extending the duration of the intervention may increase the effects [30]. Similarly, C. Comeras-Chueca found a positive effect of active video games (AVG) interventions on BMI when the intervention was longer than 18 weeks through a meta-analysis, and the difference between the groups participating for less than 18/18 or more weeks was also not significant [43].

In this light, it is important to not only study the effect on body composition of serious games but also research the method by which serious games have their effect, which would be an accelerator for better design and wider acceptance of serious games. Physical activity and dietary change are promising paths to reduce the obesity of children and adolescents.

Serious games have shown significant improvements in physical activity within children and adolescents, in line with previous studies [44,45,46]. Our subgroup analysis based on age indicates that for children and adolescents, serious games had better effect on obesity control for those under 14. We also attempted to examine the effect by socio-economic status (SES) but were impeded by the lack of detailed socio-economic data in many of the studies reviewed. However, a previous study showed that school-based health promotion has sustainable effects on remission and incidence of overweight, which were most pronounced in students of high-SES families [47]. One study showed that Dance Dance Revolution (DDR) increased vigorous physical activity [48] and helped to slow the decline in moderate-to-vigorous physical activity over time [49]. AVG interventions could significantly improve motor competence and physical activity [32,33,40,50]. Coaching is a beneficial factor for DDR use in the teenage population [31]. Higher serious game exposure and gaming progress was associated with increased frequency of physical activity [36]. However, C.B. Oliveira et al. found through a meta-analysis that AVGs were more effective than the control group in reducing BMI and body weight but not for increasing physical activity in children aged from 2 to 19 years, which is different from our research [51]. The reason might be the different types of intervention. School-based interventions were excluded from their review, while our research includes both AVGs and school-based serious games.

Diversifying the application of health education games is a promising trend [52]. As indicated in other studies, serious game intervention could increase the consumption of fruit and vegetables [11], improving healthy eating behavior [53]. Previous studies found that the effects of serious games were the greatest on knowledge [14]. One study proved that a 3-year longitudinal serious game intervention could increase adherence to the Mediterranean diet [54]. Likewise, some studies found that the Adolescent Food Habits Checklist (AFHC) was a significant mediator of the treatment effect on the BMI z-score [22].

The mechanism behind the observed effect of serious games could be the increased motivation and engagement [44,55]. Spook J. et al. found that active users’ snack consumption was found to decrease more strongly than that of inactive users [34]. A key obstacle to the improvement of body composition is lack of motivation, which can be divided into extrinsic motivation (i.e., points, achievements, or leaderboard) and intrinsic motivation (i.e., team-based competition, mastery, or autonomy) [56,57]. Some studies indicated that some game features are associated with higher user engagement, such as attractive storyline, adaptability to gender and age, high-end realistic graphics, well-defined instructions, and clear feedback and immersion [35,58]. Food aesthetics were important for most students’ fruit and vegetable intake [59]. Some participatory design and co-design games invited children and young people to become design partners, which greatly affected actual user involvement [60]. These game features should be considered in terms of game design.

Our findings emphasize the importance of sustained engagement in serious games for adolescent obesity management, highlighting that notable adiposity reduction was seen in participants adhering to intervention protocols [26] or when paired with additional weight management efforts [21]. Notably, competitive games outperformed cooperative ones in promoting adolescent weight loss [24]. When integrated into existing school or community programs, however, these interventions may face challenges such as ensuring equitable access to the necessary technology, time pressure of the school curriculum, cultural contexts (studies conducted in the United States had better outcomes, *p* < 0.01), and maintaining engagement over time (in our literature review, loss to follow-up rates were up to 66%). To address these issues, research has shown that game-based IoT solutions utilizing mobile apps and accelerometers are particularly effective [61]. Enhancing long-term engagement requires features such as adaptive challenges, personalized feedback, and competitive dynamics, complemented by immersive and social functionalities. Developers should prioritize user-centered design, collaborate with healthcare experts and the target demographic, and anchor game mechanics in behavior-change theories for maximal impact.

## 5. Limitations

This study has limitations that need to be addressed. Considerable heterogeneity was identified in the pooled results of body composition and physical activity. This heterogeneity could be attributed to various factors, including the diverse nature of the serious game interventions (such as different types of games) and the baseline characteristics of the participants (such as the SES). These elements, inherent to the study design and participant diversity, are difficult to control. In an attempt to understand these variations, our endeavors to use meta-regression to explore the heterogeneity were unsuccessful. Therefore, a series of subgroup analysis were conducted based on baseline weight status, duration of intervention, age, and country. Compared to meta-regression, subgroup analysis may not fully capture the potential interactions between factors. We attempted to examine the effect by SES but were impeded by the lack of detailed socio-economic data in many of the studies reviewed, therefore limiting our ability to perform subgroup analyses in that regard. We acknowledge that excluding non-English articles introduces a linguistic bias and may limit the scope of our findings by omitting relevant research.

## 6. Conclusions

In summary, the results of this study indicated that serious games have the potential to be effective tools for overweight/obesity control among children and adolescents. The direct effect of serious games on the increase of physical activity was observed, although a relatively long duration of trials might be needed for a sustained effect on obesity control. Dietary improvement, which could be another underlying mechanism, did not change significantly by serious game interventions. Given their potential in encouraging healthy behaviors, serious game applications should be meticulously crafted with a focus on engagement and sustainability. Such design considerations are key to maximizing their impact on weight management during the pivotal stages of childhood and adolescence.

## Figures and Tables

**Figure 1 nutrients-16-01290-f001:**
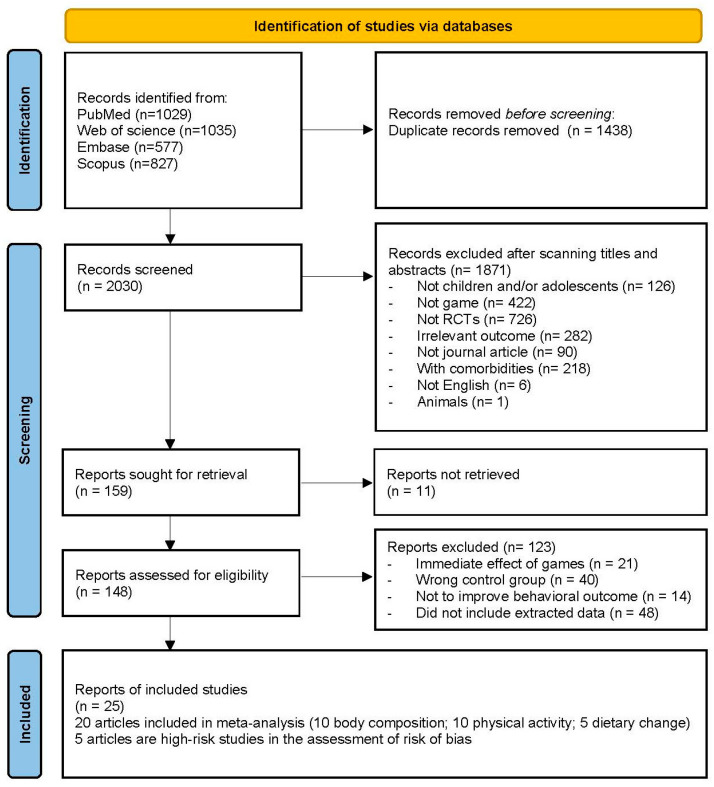
Flow diagram of the publication search process.

**Figure 2 nutrients-16-01290-f002:**
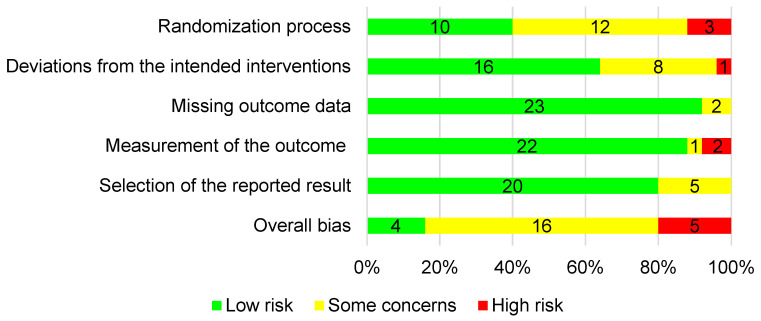
Quality assessment for the included studies in each risk-of-bias domain.

**Figure 3 nutrients-16-01290-f003:**
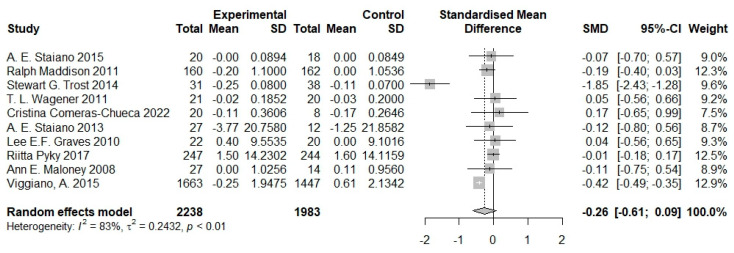
Pooled effects of serious games on body composition metrics of children and adolescents [20,21,22,23,24,25,26,27,28,29].

**Figure 4 nutrients-16-01290-f004:**
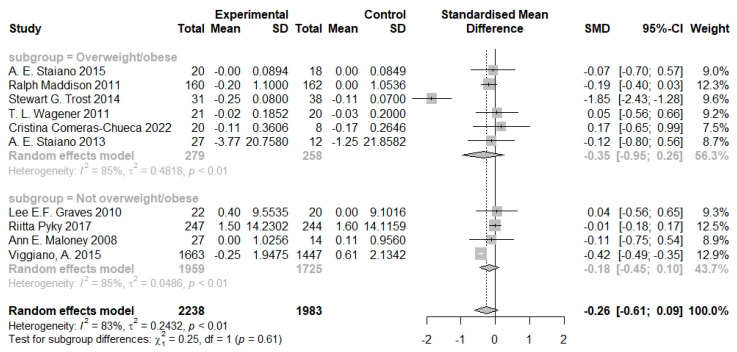
Pooled effects of serious games on body composition by baseline weight status [20,21,22,23,24,25,26,27,28,29].

**Figure 5 nutrients-16-01290-f005:**
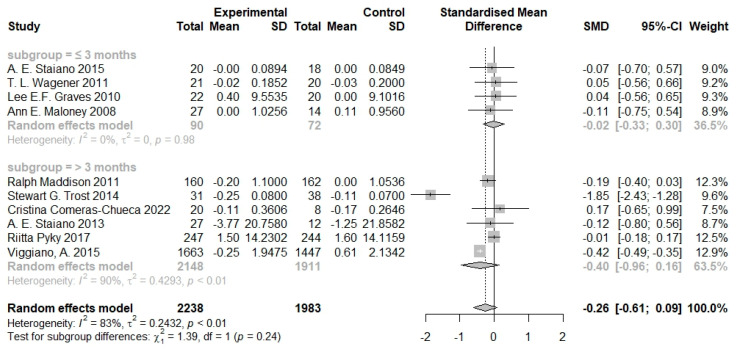
Pooled effects of serious games on body composition by duration of intervention [20,21,22,23,24,25,26,27,28,29].

**Figure 6 nutrients-16-01290-f006:**
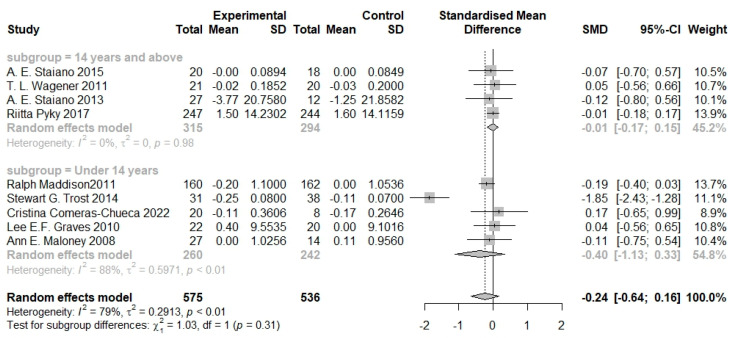
Pooled effects of serious games on body composition by age [20,21,22,23,24,25,26,27,28,29].

**Figure 7 nutrients-16-01290-f007:**
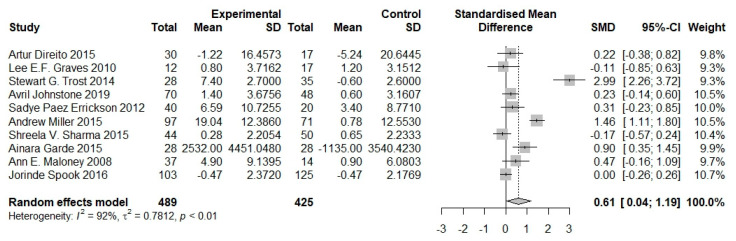
Pooled effects of serious games on physical activity of children and adolescents [21,23,29,30,31,32,33,34,35,36].

**Figure 8 nutrients-16-01290-f008:**
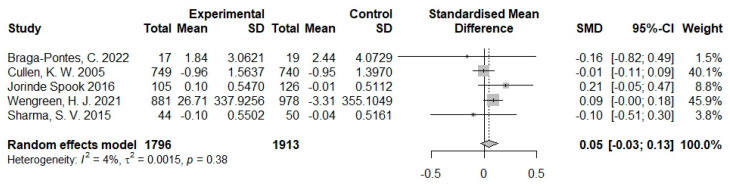
Pooled effects of serious games on dietary change of children and adolescents [34,36,37,38,39].

## Data Availability

Data used for the analysis are available from the corresponding authors on request.

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
