# Peer review of "Impact of Serious Games on Body Composition, Physical Activity, and Dietary Change in Children and Adolescents: A Systematic Review and Meta-Analysis of Randomized Controlled Trials"

_nutrients, 2024, doi:10.3390/nu16091290_

Round 1
Reviewer 1 Report
Comments and Suggestions for Authors
The main aim of this study was to conduct a systematic review and meta-analysis that investigated the impact of severe games on body composition, physical activity, and dietary change. The study showcases serious games as a potential tool for increasing physical activity in children and adolescents, which may help in tackling obesity. Although the results on body composition and diet were modest, the games demonstrated notable potential in enhancing physical activity, implying that longer interventions could lead to improved outcomes.
Here are some potential limitations that authors may address in the text:
Introduction
It would be beneficial to detail more specifically how serious games affect physical activity and dietary change.
The introduction mentions the limitations of previous meta-analytical studies. However, it does need to specify how this study overcomes those limitations, in addition to conducting a more up-to-date review.
Improving the justification for the realization of this study.
While the introduction ends with a statement of objectives, it could be more specific regarding the hypotheses or research questions.
Despite introducing serious games as a potential solution, a discussion on the existing challenges and limitations of this type of intervention is lacking.
Methodology
The methodology described for the study follows the PRISMA guidelines.
Given the scarcity of studies, excluding non-English articles may limit the scope of the study by omitting relevant research published in other languages, potentially introducing a linguistic bias.
Results
The presentation of the study's results is detailed and includes a variety of analyses.
Discussion
A more detailed and critical analysis of methodological differences or intervention designs that could explain discrepancies in the results would be valuable.
The discussion acknowledges the heterogeneity among the studies but could delve deeper into possible underlying reasons for this variation in addition to those already mentioned. Factors such as participant age, cultural contexts, and specificities of the games could be examined more thoroughly.
How can these interventions be integrated into existing school or community programs? What are the challenges? These are important aspects of the practical implications that deserve discussion.
It suggests that longer interventions might be necessary for sustainable effects. However, it is essential to specify what strategies can be employed to keep children motivated to use serious games over the long term. Additionally, within this question, what specific features of the games are most effective? How can game developers incorporate these insights into the design of new games?
The mentioned limitations, such as the heterogeneity of the studies and the lack of socioeconomic data, are important. However, the discussion could benefit from a deeper reflection on how these limitations affect the interpretation of the results and the confidence in the conclusions. For instance, how might the diversity in the types of games and the characteristics of the participants have influenced the outcomes? Are there other methodological limitations that were not addressed?
The principal critique of this study is the superficial analysis of limitations and the underestimation of practical implications without an in-depth exploration of how these factors influence the applicability of serious games in the control of childhood obesity.
Author Response
Please see in the attachment.

Reviewer 2 Report
Comments and Suggestions for Authors
Dear Authors,
I thank the Editor for entrusting me to review this manuscript. As the authors of the manuscript write, obesity among children has been steadily increasing over the past few decades. The problem affects populations worldwide and all age groups. Various intervention strategies are being introduced to reduce this problem. One such modern solution is serious games, which aim to promote a healthy lifestyle through proper diet and physical activity. I congratulate the authors of this study for taking on this topic. The authors, using advanced analytical techniques, using the results of published materials available in electronic databases, conducted a meta-analysis.
The study designed and carried out correctly. Described in detail the process of selection of research material. The presented results and their analysis are not questionable.
Author Response
We are really grateful that the reviewer took the time to read our manuscript and showed positive evaluation of the work. Your valuable evaluation of our work is highly appreciated, and we are delighted that you have acknowledged and recognized our efforts.
Reviewer 3 Report
Comments and Suggestions for Authors
The study presented here addresses the very important issue of assessing the effect of serious games on weight reduction in children. Subgroup analyses revealed important differences in the context of intervention planning: Firstly, overweight or obese children and adolescents showed a more pronounced improvement in body composition compared to normal weight peers. Secondly, relatively long-term serious games interventions showed a more significant change in body composition than short-term interventions.
This synthetic study of the relationship between weight change in overweight children and adolescents in the context of serious games provided guidance on how to plan an intervention to be effective and on which elements to pay attention to (duration of exercise, intensity, type, eating habits) in order to obtain the best possible weight loss results.
Chapter 5 should also describe the strengths of the systematic review presented. Perhaps the passage "Compared to previous studies, our research is the first systematic review and meta- analysis of RCTs to comprehensively estimate...". should be moved to Chapter 5 entitled Strengths and limitation.
Author Response
We are really grateful that the reviewer took the time to read our manuscript and showed positive evaluation of the work.
Due to the formatting requirements of Nutrients, we regret that we may not be able to include a dedicated section for "Strengths and Limitations" in Chapter 5. Instead, we revised the description in our manuscript to specify how we overcome the limitations of previous studies in more detail (Page 2, Line 77-83).
Thank you for taking the time to provide us with feedback. We are genuinely delighted that our efforts have been recognized and acknowledged.